# Assessing the potential of seaweed extracts to improve vegetative, physiological and berry quality parameters in *Vitis vinifera* cv. Chardonnay under cool climatic conditions

Johan Yssel [1,2,3], Vicky Everaerts[4], Wendy Van Hemelrijk[4], Dany Bylemans[2,4], Mathabatha Evodia Setati [3], Bart Lievens[1,2], Erna Blancquaert[3], Sam Crauwels [1,2]*

**1** Department of Microbial and Molecular Systems (M2S), CMPG Laboratory for Process Microbial Ecology and Bioinspirational Management (PME&BIM), KU Leuven, Leuven, Belgium, **2** Leuven Plant Institute (LPI), KU Leuven, Leuven, Belgium, **3** South African Grape and Wine Research Institute, Stellenbosch University, Stellenbosch, South Africa, **4** Research Station for Fruit Growing, Sint-Truiden, Belgium

\* sam.crauwels@kuleuven.be

## Abstract

Seaweed extracts are promising plant biostimulants for viticulture, but their effects on white winegrape cultivars grown under cool climates remain fairly undocumented. Furthermore, information is limited on the biostimulant potential of some brown seaweed species like *Ecklonia maxima*. This study evaluated the impact of two commercial extracts (derived from *Ascophyllum nodosum* and *Ecklonia maxima*) on *Vitis vinifera* cv. Chardonnay in Belgium during the 2021 and 2022 growing seasons. The extracts, alongside a water-control and an NPK-reference (NPK-Ref) treatment (with nitrogen, phosphorus, and potassium levels comparable to the extracts), were applied as foliar sprays five times at regular intervals, from flowering to ripening. In 2021 and 2022, *A. nodosum* significantly increased individual leaf area (+12% and +15%), while in 2021 *A. nodosum*-treated vines had an increased chlorophyll content index (+12% CCI) and photosystem II (PSII) reaction centre density (+6%) relative to control vines. This corresponded with a small, but significant, improvement (+1.5%) in PSII maximum quantum yield ($F_v/F_m$), whereas PSII electron transport efficiency ($\Phi_{E0}$) remained unchanged. Furthermore, increased berry size, mass, and sugar content were observed in *A. nodosum*-treated vines during ripening in 2022, comparable to NPK-Ref vines. Conversely, the *E. maxima* extract had limited effects on vegetative growth, physiology, and subsequent berry development. Yield increase from 2021 to 2022 varied by treatment, with a significant increase observed for *E. maxima* (+60%) and NPK-Ref vines (+80%), relative to control vines. Our results indicate that seaweed extracts, specifically *A. nodosum*-based, can enhance grapevine leaf area, CCI, and $F_v/F_m$ under cool climatic conditions. *A. nodosum* treatment was also associated with increased berry size and sugar content, while *E. maxima* treatment was

**Data availability statement:** All relevant data are within the manuscript and its Supporting Information files.

**Funding:** Johan (DJ) Yssel Project number: 3E200528 KU Leuven Campus Group T https://www.kuleuven.be/english/campuses/group-t-leuven-campus/Research The sponsors played no additional roles apart from financing.

**Competing interests:** The authors have declared that no competing interests exist.

associated with increased yield in the subsequent, warmer season. Altogether, our study highlights that the differential effects of seaweed extracts on grapevine development are modulated by species and environmental conditions.

## Introduction

Grapevine is one of the most economically valuable crops in the world, with grapes being the fifth most produced fruit crop globally [1]. Approximately 46% of grapes produced are used for winemaking [2]. Grapevines are cultivated under diverse environmental conditions, with regions differing considerably in terms of growing season temperature, sunlight exposure, rainfall and humidity, wind, elevation, and soil characteristics [3]. Currently, most of the world's wine production occurs in the mid-latitudes, where the average growing season temperatures vary between 13 and 22 °C [4,5]. However, climate change is expected to reduce the viticultural suitability of many established winemaking regions [6–8]. In fact, 90% of traditional wine regions in coastal and lowland regions of Spain, Italy, Greece and southern California could be at risk of disappearing by the end of the century [9] due to increased heat- and drought stress and changes in precipitation patterns [10–12], as well as the spread of new pests and rise in disease incidence [13]. Altogether, these challenges will result in reduced yields and alterations in grape quality, leading to lower wine quality and/or production [14,15]. Nevertheless, while certain areas are under threat, opportunities arise as new production regions emerge in the cooler regions of North-western Europe, namely Belgium, northern Germany, the Netherlands, and the United Kingdom [9].

In response to the challenges posed by climate change, growers are increasingly relying on (chemical) fertilisers and pesticides as measures to ensure the consistent production of high-quality grapes [16,17]. However, researchers have reported that the excessive application of macronutrients such as nitrogen and phosphorous leads to reduced soil health and can inadvertently favour the development of pathogens over beneficial microorganisms [18–20], thereby negatively impacting the wine quality [21]. Likewise, excessive pesticide use has been linked with surface water contamination, reduced soil health and biodiversity [22,23], as well as human health consequences [24,25]. Thus, there is an urgent need for alternative strategies to ensure the long-term sustainability of viticulture [26–28].

The use of biostimulants has emerged as a sustainable solution to improve vineyard resilience, offering the potential to enhance plant growth, performance, stress tolerance, and crop quality, while reducing the need for synthetic agrochemical inputs [29–31]. Biostimulants have long been loosely defined and were often regarded dubiously because of their composite nature and the inherent difficulty to determine which specific components were making beneficial contributions. The definition proposed by du Jardin [32] which stipulates biostimulants as "any substance or microorganism applied to plants with the aim to enhance nutrition efficiency, abiotic stress tolerance and/or crop quality traits, regardless of its nutrients content" represents one of

the clearer and more concise ways to define biostimulants. Biostimulants can be classified in several ways, for example based on mode of action, composition, or source origin [33]. Following the source origin approach by Yakhin [33], biostimulants are classified into seven categories, including humic substances, protein hydrolysates, seaweed extracts, chitosan and other biopolymers, inorganic compounds, beneficial fungi, and beneficial bacteria. Of these, seaweed extracts, in particular those made from brown seaweed species such as *Laminaria, Sargassum* and *Ascophyllum* [34], have shown promising features in horticulture [35] as well as viticulture [21,36]. They contain various beneficial minerals, organic acids, amino acids, and plant growth regulators such as cytokinins and auxins [37–39], contributing to improved stress tolerance and crop productivity and quality [35,40–43].

Seaweed extracts have shown stress-ameliorative effects in grapevines, such as improving stomatal conductance and maximum quantum yield of photosystem II (PSII) ($F_v/F_m$) under drought stress [44,45] and combined light and temperature stress [45]. However, research is still limited on their application and effects in commercial winegrape varieties, particularly white winegrape varieties [21,46]. Furthermore, while most studies have focused on the effects of seaweed extracts in drought and heat-prone climates [44], only a limited number of studies have been conducted on their effects in cooler climatic regions [21]. These cooler regions are challenged by variable weather conditions and increasingly by extreme weather events, such as seasonal heat stress or drought events, to which vines in these regions are less adapted than their counterparts in warm climates [47]. In contrast to warm-climate regions, cooler, humid regions are more susceptible to delayed ripening or increased disease incidence. In these environments, seaweed extracts may stimulate natural processes that enhance nutrient uptake and use efficiency, maintain photosynthetic efficiency, and increase tolerance to abiotic stressors [48]. Vines that are well fertilised and in a good physiological condition are generally better prepared to respond to biotic and abiotic stressors. However, the physiological benefits conferred by seaweed extracts are likely influenced by the specific composition of the extract, which can vary substantially depending on species, production process, and season. For example, protein fractions can range from 3–15% and polysaccharides, a major constituent, from 30–40% [35]. One study reported that for *A. nodosum*, total polyphenol content differed by 40% and fucoidan content by 20% between two seasons [49].

The aim of this study was to assess whether the foliar application of seaweed extracts leads to improvements in vegetative, physiological, and berry quality parameters of Chardonnay vines grown in a cool climate region (Belgium). Specifically, two commercial extracts from brown seaweed species, *Ascophyllum nodosum* and *Ecklonia maxima*, were evaluated alongside a water treatment (negative control) and an NPK-reference (NPK-Ref) treatment containing comparable concentrations of the macronutrients nitrogen, phosphorus, and potassium, as found within the seaweed extracts. *Ascophyllum nodosum* is an important species of brown seaweed, which grows in the intertidal region of the coasts of the North Atlantic, and has demonstrated significant plant growth promoting and abiotic stress protective effects in several agricultural crops [48,50,51] as well as several winegrape cultivars. For example, in a hot summer climate region in Australia, the soil application of *A.nodosum* resulted in increased Chardonnay grape yield [21,52]. In contrast, *E. maxima*, a species of brown seaweed found in the shallow, temperate regions of the Atlantic coast of Southern Africa, has remained understudied regarding its effects on grapevine vegetative growth and physiology. Nevertheless, the limited number of studies suggest that *E. maxima* can positively influence berry quality parameters [53,54]. *Ecklonia maxima* shows particular promise to be commonly used as a biostimulant as it grows over a wide range of temperatures [55] and, in contrast to some seaweed species, has exhibited a range expansion, partly due to localised nearshore upwelling that brings cooler, nutrient-rich water to the surface [56,57]. To evaluate the growth stimulating effects of these seaweed extracts under cool climatic conditions and how these effects may vary across different seasons, our study was conducted in Belgium, an emerging wine region with a cool climate, over two growing seasons (2021 and 2022). Two different brown seaweed species were included to determine if any observed positive effects are dependent on the seaweed species used. The findings of this research provide valuable insights into the application of seaweed extracts, to enhance cool climate

viticulture practices, improve vine growth, and grape ripening dynamics in terms of total soluble solids and acid evolution, and advance sustainable agriculture in white winegrape production.

## Materials and methods

### Vineyard layout

The study was conducted during two consecutive growing seasons (2021 and 2022) on the experimental vineyard of Research Station for Fruit Growing, located 5 km south of the town Sint-Truiden, Belgium (50.774218 N, 5.155002 E) at 75 m above sea level. A total of 120 six-year-old vines of *Vitis vinifera* cv. Chardonnay (clone ENTAV-INRA 96), grafted onto SO4 rootstocks (*Vitis berlandieri* x *Vitis riparia*) were included in the study. Vines were planted in sandy-loam soil in a single row within the vineyard, which in total had 22 rows of vines. Row orientation was NNW/SSE, with row and vine spacing 2.0 m and 1.0 m, respectively (S1A Fig). The vines were trained on a six-wire vertical shoot positioning (VSP) and cane-pruned single guyot system with two spurs on alternating sides of the crown. On alternating years, a fruiting cane and a backup cane was trimmed down to eight nodes, while on the opposing side the spur was trimmed down to two nodes from which the shoots used in the subsequent year were grown. Canopy management involved minimal leaf removal in the fruiting zone and two instances of topping (S1B Fig).

### Experimental design

Vines were divided into four treatment groups consisting of 12 vines each, according to a randomised block design. All four treatments were prepared in demineralised water at a 1:500 dilution with 0.001% Tween-20 (Merck KGaA, Darmstadt, Germany) added as a surfactant to improve foliar uptake of the treatments, according to supplier specifications. The treatments were (i) a blank treatment containing only demineralised water and Tween-20 with no additional compounds (Control), (ii) a reference treatment for macronutrient content (NPK-Ref); (iii) a commercial *A. nodosum* based seaweed extract (Acadian Plant Health, Dartmouth, Canada); and (iv) a commercial *E. maxima*-based seaweed extract (COMPO EXPERT GmbH, Münster, Germany) (S2A Fig). NPK-Ref was prepared from a mixture of KCl (Honeywell International Inc, Charlotte, NC) and commercial monoammonium phosphate (Haifa MAP™ Haifa Group, Haifa, Israel) to approximate the maximum nitrogen, phosphorus, and potassium content (NPK) in the two biostimulant treatments. While it was possible to mimic the maximum N and K in the extracts, the P-content was 67% more than that of *E. maxima*, which contained the highest P of the extracts (S1 Table). The Soil Service of Belgium (Leuven, Belgium) determined the nutrient composition of the seaweed extracts. The contents of P, K, Mg, Ca, Na, S were analysed by inductively coupled plasma atomic emission spectroscopy (ICP-AES), while total N and total C were determined using spectrometry, according to standard protocols, following the Belgian Accreditation Institution (BELAC) reference methods. Treatments were applied at five specific phenological development stages once vegetation had actively started developing, based on supplier recommendations to enhance fruit sizing. In season 2021 applications commenced every three weeks, starting at 18 days before flowering (11 June), which corresponded to stage E-L 15 according to the modified Eichhorn-Lorenz (E-L) system [58], and every two weeks during season 2022, starting 12 days before the flowering stage (E-L 18/19) (3 June). The final treatment was applied before véraison (E-L 34) in both seasons(3 September 2021 and 5 August 2022) (S2A Fig). To ensure consistency in plant response, the same phenological stages were targeted in both seasons. As a consequence, differences in application dates and frequency arose, with shorter intervals between treatments in 2022 due to faster phenological progression under warmer climate conditions, as detailed in the seasonal weather results section. On alternating weeks, the vines were treated with potassium hydrogen carbonate (Karma®, Certis Europe B.V, Brussels, Belgium) as a minimal intervention approach to reduce the pressure of *Erysiphe necator* (the causal agent of powdery mildew in grapevines) (S2A Fig). Treatments were applied in four blocks, each consisting of 30 vines, to a specific subblock consisting of five vines. Additionally, each block contained a subblock of five vines that remained untreated. Between subblocks there was another

vine that remained untreated as a buffer between subblocks (S2B Fig). Treatments were applied by spraying both sides of the canopy using a STIHL SR 430 gas-powered backpack sprayer (ANDREAS STIHL NV, Puurs-Sint-Amands, Belgium) until run-off. All sprays were administered between 8:00 and 9:00 in the morning on clear days. Samples were collected from, and measurements done on, the three inner vines of each treated subblock (3 plants × 4 blocks per treatment), to minimise the impact of treatment carryover between subblocks (S2B Fig).

## Weather data

Environmental conditions were logged via an automated weather station located in the vicinity of the vineyard. The daily maximum ($T_{max}$), average ($T_{avg}$) and minimum temperature ($T_{min}$), precipitation, and relative humidity (RH%) were logged from 1 April to 31 October during the 2021 and 2022 seasons. Cumulative growing degree days (GDD) [59] were calculated using the following formula:

$$GDD = \frac{1}{n} \sum_{i=0}^{n} (T - T_b)$$

(1)

where n is the number of days, T the daily mean temperature and $T_b$ the base temperature (10 °C) for grapevine growth. Vapour pressure deficit (VPD) was calculated hourly using the following formula proposed by [60]:

$$VPD = 0.61078 \times e^{\frac{17.2694T}{(T+237.3)}} \times \frac{1-RH\%}{100}$$

(2)

where VPD is in kPa, T is the temperature in °C, and RH% the relative humidity. VPD data were then plotted over the course of the growing season.

## Vine vegetative characteristics and leaf starch content

Three representative shoots per vine were tagged on the main fruiting cane at E-L 12 in the 2021 growing season (21 May 2021). These were measured weekly over three weeks with a measuring tape, after which the average shoot length was determined for each replicate vine. Treatments were then assigned to subblocks to ensure each treatment had vines of a comparable phenological development, with each sampled vine a replicate (n = 12) (S2B Fig). During the 2022 growing season three new representative shoots were tagged on the fruiting cane of vines (receiving the same treatments as in 2021) and measured weekly starting at E-L 15 (18 May 2022) until one week before the start of flowering (E-L 21). Following the start of treatment application (E-L 15 on 11 June 2021 and E-L 18/19 on 3 June 2022), leaf destructive measurements were done every 2-3 weeks on a single representative, fully expanded mature leaf, exposed to sunlight, which was sampled from each vine (n = 12). The sampled leaves were kept on ice during sampling in the field and stored in the laboratory at -20 °C until analysis. Leaves were scanned in colour at 400 dpi using a Ricoh IM C4500 multi-function printer (Ricoh, Vilvoorde, Belgium), and the leaf area of individual leaves (hereafter referred to as 'leaf area') was determined using the Fiji software package [61]. After scanning, leaf petioles were removed with a scalpel and the fresh weight was recorded using an analytical balance (PIONEER PX 323, Ohaus, Nänikon, Switzerland). Leaves were then dried at 70 °C until a constant dry mass was achieved to calculate the leaf water content, dry matter percentage (DM%), and specific leaf area (SLA) by dividing the leaf area by the leaf dry mass.

Each dried leaf per replicate vine (n = 12) was ground to a powder using a mortar and pestle, and the soluble sugars were extracted using an ethanol extraction step, after which the leaf total starch content was determined using a commercial kit for total starches, K-TSTA-100 (Megazyme International, Bray, Ireland). Firstly, the soluble sugars extraction was done according to [62], with the following modifications: for the ethanol extraction, 400 µL of 80% ethanol was added to 50 mg of dried leaf powder in a 2 mL screw cap tube. Then, samples were vortexed and boiled for 15 minutes at 80 °C in

a heating block (Model 2053-1, Lab-Line Instruments Inc., Melrose Park, IL). Samples were then vortexed again and centrifuged at 12 500 RCF for 10 minutes and the supernatant was discarded, retaining the leaf powder pellet. These steps (adding ethanol to the pellet, vortexing, centrifuging, and discarding the supernatant) were repeated another two times to ensure most of the soluble sugars were removed. Then the remaining pellet was dried overnight at 70 °C and the starches were extracted using the aforementioned starch extraction kit (with the protocol optimised for use with microtiter plates [63]). Briefly, this involved adding 100 μL of 80% ethanol and 250 μL of ice-cold 1.7 M NaOH (VWR, Leuven, Belgium) and two 2 mm glass beads to the screw cap tube containing the dried leaf pellet. Tubes were then vortexed, and the content was homogenised in a bead mill homogeniser (Bead Ruptor Elite, Omni International, GA, USA) using a speed of 4.5 m s$^{-1}$ with five cycles of 30 seconds each. Next, 1000 μL of a 100 mM sodium-acetate buffer at pH 3.7 was added and the tubes were vortexed. Then, 12.5 μL of undiluted thermostable α-amylase was added, followed immediately by 12.5 μL amyloglucosidase, after which the tubes were incubated at 50 °C for 60 minutes to break down and convert the starch to D-glucose. Tubes were then centrifuged at 13 000 RCF for three minutes and the retained supernatant transferred to new 1.5 mL microcentrifuge tubes. The sample supernatants were then diluted with 100 mM sodium-acetate buffer at pH 5.0, depending on the expected final concentrations of starch (e.g., 1:4 dilution early season and 1:7 late season). Subsequently, following brief vortexing, 20 μL of each sample was transferred in duplicate to a flat bottom 96-well plate (96 MicroWell™ Plates, Nunc™, VWR, Leuven, Belgium), to which 300 μL glucose oxidase/peroxidase (GOPOD) reagent (from the K-TSTA-100A extraction kit) was added. Plates were closed and incubated at 50 °C for 20 minutes, after which the absorbance was read using a SpectraMax® ABS spectrophotometer (Molecular Devices, San Jose, CA, USA) with the wavelength set to 510 nm. The total starch was then calculated using a standard curve of glucose as specified in the kit protocol.

## Chlorophyll fluorescence indices and stomatal conductance

Two mature, fully expanded leaves, exposed to full sunlight, and of a similar age (corresponding to leaves between the third and fifth node of the shoots) were tagged per replicate vine ($n = 12$) for leaf physiological measurements. The leaf chlorophyll content index (CCI) was measured in duplicate on each tagged leaf (giving four readings that were averaged per replicate vine) using a CCM-200 chlorophyll content meter (Opti-Sciences, Hudson, NH) between 10:00 and 11:30 in the morning. The CCI is a transmittance-based measure of the relative chlorophyll content in leaves with a species-dependent power-law relationship with respect to absolute chlorophyll concentration [64]. Next, on the same leaves and within the same time interval, the dark-adapted fast chlorophyll-*a* fluorescence induction (OJIP) transient was measured and recorded using a handheld Pocket PEA chlorophyll fluorimeter (Hansatech Instruments, Norfolk, United Kingdom). From the OJIP induction curves the following photosystem II (PSII) parameters were derived: the reaction centra density per excited cross section (RC/CS), the maximum quantum yield of PSII photochemistry ($F_v/F_m$)—often used as a measure of plant performance or photosynthetic potential [65], and the quantum yield for electron transport beyond QA$^-$ (PSII electron transport efficiency, $\Phi_{E0}$). For reference, in healthy, unstressed leaves, the maximum quantum yield ($F_v/F_m$) typically approaches 0.83 [66]. Before measuring, leaves were dark-adapted for 30 minutes to ensure all PSII RCs were open, allowing for estimation of the minimal fluorescence ($F_0$). Leaves were then subjected to a saturating actinic light flash (627 nm, 3500 μmol m$^{-2}$ s$^{-1}$), to determine maximum fluorescence ($F_m$). The maximum quantum yield of PSII photochemistry ($F_v/F_m$) was calculated as ($F_m$–$F_0$)/$F_m$ [67]. The two leaf readings were averaged per replicate vine.

Additionally, leaf stomatal conductance was measured between 11:30 and 14:00 on the abaxial (under) side of the first tagged leaf (as measured from the crown of the vine) for each replicate vine ($n = 12$). Measurements were done for 30 seconds until steady state using an SC-1 leaf porometer (METER Group, Pullman, WA, USA). Parameters were measured weekly (between 15 June and 25 August in 2021; and 15 June to 12 August in 2022), except for days with rain or full cloud cover, where the measurement was taken the following week on a cloudless day.

## Berry classical parameters and morphology

Berries were sampled from véraison (E-L 34) until harvest (E-L 38) every 14 days for the 2021 growing season, and every 10-14 days for the 2022 season, with delayed sampling of at least two days following rainfall to reduce the impact of sugars dilution (S2C Fig). A total of about 80 representative berries were sampled from healthy clusters across the inner three vines of a treated panel (subblock) ($n=4$), using sterile nitrile gloves, into resealable bags, sampling from the top, middle, and bottom of the cluster, and alternating between sun-exposed and shaded sides. Berries were kept on ice during sampling and transport until further processing. Thirty representative berries that were used for morphological measurements ($n=4$) were stored in resealable bags at -20 °C, while the remainder were immediately processed at the laboratory for juice analyses. Briefly, the remaining berries were gently crushed by hand, externally through the bag, to extract a representative juice sample for each subblock ($n=4$), and the juice was decanted and centrifuged at 4000 RCM for five minutes. The clear must was used to determine total soluble solids (TSS in °Brix), pH, and titratable acidity (TA in g L$^{-1}$). TSS was determined with a digital refractometer (HI 96801, Hanna Instruments, Temse, Belgium) in duplicate and averaged per replicate. A pH probe (HI 1131, Hanna Instruments, Temse, Belgium) was used to determine the pH, while TA was measured using the same probe in conjunction with an autotitrator (HI 902, Hanna Instruments, Temse, Belgium). Each sample consisted of 20 mL juice and 20 mL demineralised water, titrated with 0.33 M NaOH (VWR, Leuven, Belgium) to a pH of 7.01, with results reported as tartaric acid equivalent in g L$^{-1}$. For berry morphological measurements, 20 undamaged berries were randomly selected from each sampled subblock ($n=4$). The berry length from pedicel to the bottom and berry diameter were measured using a vernier calliper (HOLEX 412821_100, Hoffman Group, Borne, The Netherlands). Individual berry weights were also recorded on an analytical balance (PIONEER PX 323, Ohaus, Nänikon, Switzerland), taking care that the berries remained frozen and surface ice was removed. The berry volume was calculated from the berry length and diameter, assuming a perfect spherical shape. Individual berry densities were calculated by dividing the berry mass by the berry volume. Following this, the 20 berries were dried at 75 °C until a constant mass was observed to determine the berry dry mass and DM%. The morphological measurements of the 20 berries were averaged per replicate subblock ($n=4$).

## Harvest data

At technological maturity (TSS = 21 °Brix) all berry clusters were harvested using sterile nitrile gloves into separate bags for each of the inner three vines per subblock, yielding 12 bags per treatment. The clusters were counted and weighed for each replicate vine ($n=12$), to obtain an average number of clusters and average cluster weight for each replicate vine. Following this, the clusters were pooled per treated block, yielding four replicates per treatment. From each replicate, 100 berries were randomly sampled, of which 30 were frozen at -20 °C for morphological measurements and the remainder were immediately crushed to juice for analysis (as described previously).

## Statistical analysis

A series of linear mixed models (LMMs) were fit against the longitudinal vegetative and berry quality data. A significance level of $\alpha=0.05$ was used to determine significant differences in all analyses. For the vegetative dataset, an initial global model was constructed using the 'lmer' function from the R lme4 package [68]. Fixed effects included season, treatment, and time (with all interactions among these factors), while environmental covariates (temperature, rain, humidity, VPD, and GDD) were incorporated as additive predictors. Each replicate vine ($n=48$ per timepoint, with 12 per treatment) was included as a random factor nested within block ($n=4$). This was done to account for the repeated measures taken on each vine and for the non-independence of observations, thereby preventing pseudoreplication [68]. To identify the most parsimonious model, the 'dredge' function from the MuMIn package was used to compare candidate models starting from the global model estimated via restricted maximum likelihood (REML), while constraining all models to include the fixed

effects season, treatment, and time. The most parsimonious model was selected based on the lowest corrected Aikaike Information Criterion (AICc) [69]. Subsequently, separate analyses were performed for each year by fitting optimal LMMs via 'dredge;' in these within-year models the design was balanced, as each treatment was associated with identical levels of time, with the same number of replicates per timepoint ($n = 12$).

In all cases, model assumptions were thoroughly checked. Specifically, we examined residual versus fitted value plots and Q-Q plots to assess normality and homogeneity of variances, while simulated residuals were generated using the DHARMa package, to further assess model fit and detect potential issues such as non-uniformity, over- or under-dispersion, and zero inflation. When evidence of heteroskedasticity was observed, models were refitted using the 'lme' function from the nlme package with an appropriate variance structure (e.g., using weights = varIdent(form = ~ 1 | Time:Treatment)) and maximum likelihood estimation (ML). These models were compared to the previous optimal LMM using AICc and $R^2$ to determine if model fit was significantly improved ($\Delta AICc > 10$) [69]. Final models were refitted using REML, and an analysis of variance (ANOVA) type III test was performed with Kenward-Roger degrees of freedom estimation to assess the significance of the fixed effects. The analysis was followed by a *post hoc* pairwise comparison using estimated marginal means (EMM) with Satterthwaite degrees of freedom adjustment, via the 'emmeans' package [70]. In cases where the model included a significant interaction term, EMMs were evaluated at each level of the interacting factor using the Šidák confidence level adjustment for multiple comparisons. In the absence of significant interaction effects, pairwise comparisons among treatment levels were conducted using EMMs with Tukey adjustment for multiple comparisons, implemented via the 'emmeans' package.

For the berry quality data, a similar modelling approach was followed with season and treatment as factors, including the aforementioned environmental covariates as additive predictors in the global starting model. The exception was that 'time' was modelled using a 2nd order polynomial of the means-centred DAAs, to account for slowing ripening dynamics over time. Furthermore, as berry sampling was done per treatment per block, the total replicates per timepoint were reduced to $n = 16$, with four per treatment. Data were first assessed for normality and homogeneity of variances. Continuous response variables (e.g., yield, average cluster mass) were analysed using LMMs, whereas count data (e.g., number of berry clusters) were modelled with a Generalised Linear Model (GLM) assuming a negative binomial distribution with a log link (using the 'glm' function from the lme4 package) [68]. Following significant associations of season or its respective interaction terms with a response variable, each season was modelled separately—with treatment, time, and their interaction as fixed effects, and block as a random factor. For each season, the most parsimonious model was selected (based on AICc) and ANOVA was conducted on that model to report overall effects. *A priori* planned pairwise comparisons were performed using EMMs [70] on the model that retained the treatment × time interaction—even if that term was excluded from the final parsimonious model. This allowed us to analyse harvest-specific contrasts rather than merely reflecting the overall treatment effect. All contrasts (including sequential differences between timepoints, the overall treatment effect, and the treatment effect at harvest) were adjusted with the Šidák correction for multiple comparisons. For yield parameters, *a priori* analyses compared the change from 2021 to 2022 within each treatment (i.e., using EMMs and excluding cross-treatment interaction contrasts). Correlations between variables were assessed via a repeated-measures correlation using the R package 'rmcorr,' to account for the repeated measures. Results were visualised using the 'ggplot2' package [71]. All analyses and visualisation of the data were performed in R version 4.2.3 [72].

## Results

### Weather data

Whereas the nine-year average of cumulative growing degree days (GDDs) between 2013 and 2021 was 1114, during the 2021 growing season (April to October) 1001 GDDs were accumulated, while 2022 saw a total of 1230 GDDs. In 2021, temperatures were cooler than average in April (-2.8 °C) and May (-0.9 °C), warmer in June (+2.0 °C), and cooler

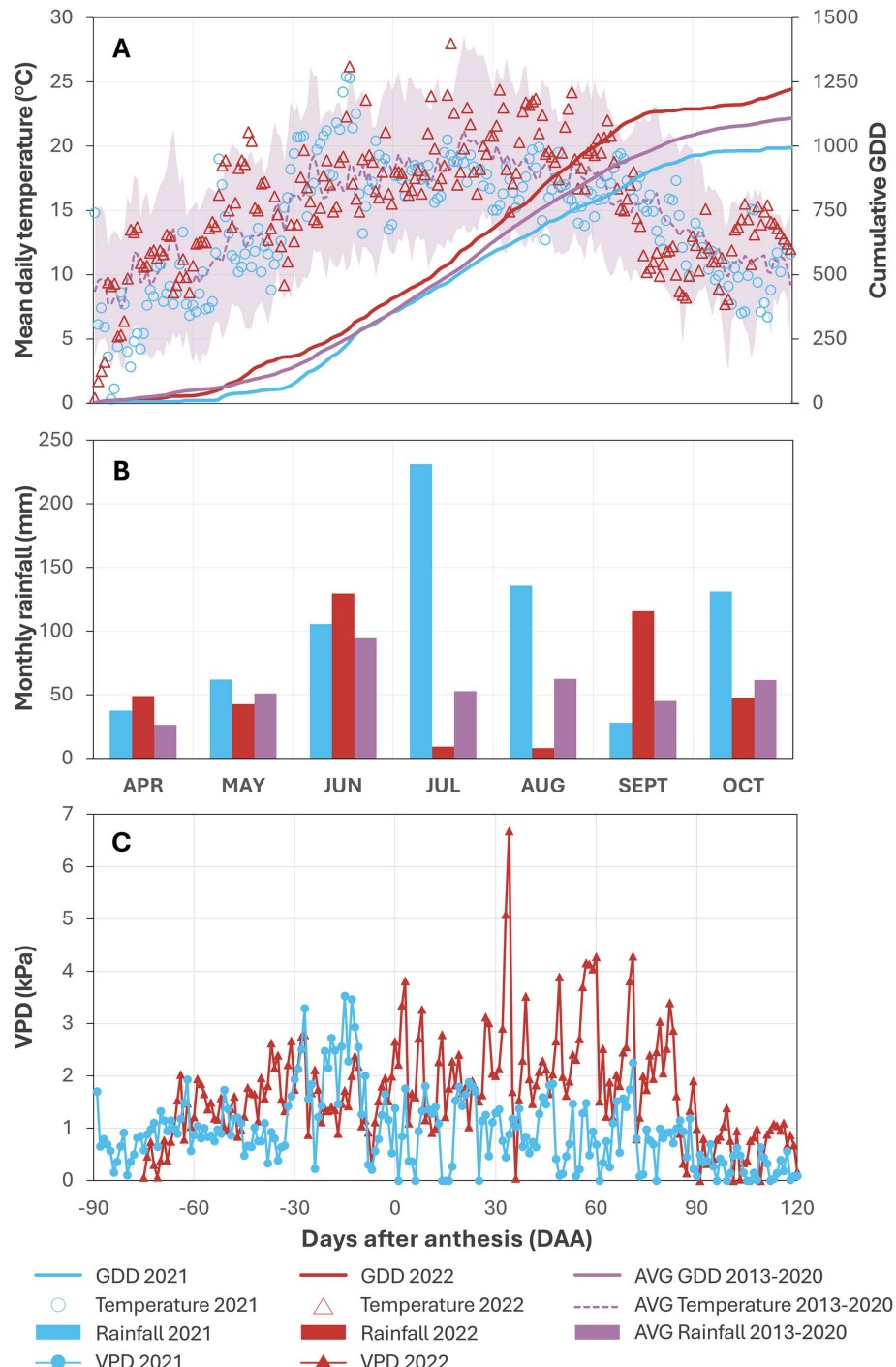

**Fig 1. (A) Mean daily temperature and cumulative growing degree days (GDD), (B) monthly rainfall, and (C) vapour pressure deficit (VPD) for the 2021 and 2022 seasons.** (A) Mean daily temperatures are indicated by blue circles for 2021, red triangles for 2022, and a purple dashed line for the average of the preceding eight years (2013-2020). The purple shading indicates the temperature range over this period, and solid lines represent the GDDs. The maximum daily VPD is shown with blue circles for 2021 and red triangles for 2022 (C).

in July (-1.9 °C) and August (-1.8 °C) (Fig 1A). Rainfall was significantly higher in July (231 mm vs 53 mm) and August (136 mm vs 62 mm), with lower-than-average rainfall in September (28 mm vs 45 mm) (Fig 1B). Vapour pressure deficit (VPD) exceeded 3 kPa only on three days before anthesis (flowering) (Fig 1C). In 2022, compared to the nine-year average (2013 to 2021) April was slightly cooler (-0.7 °C), followed by a warmer May (+2.4 °C), average June and July, and warmer August (+2.1 °C), with a cooler September (-1.1 °C) (Fig 1A). Rainfall was above average in April and June, but below average in July (9 mm vs 53 mm) and August (8 mm vs 62 mm), with higher-than-average rainfall in September (116 mm vs 45 mm) (Fig 1B). The 2022 season had elevated VPD values post-anthesis (flowering), exceeding 3 kPa on 18 days, peaking at 6.7 kPa at 34 days after anthesis (Fig 1C).

## Seaweed extract effect on grapevine vegetative growth and physiology

In 2021, vines were treated with a negative control (water), two seaweed extracts based on *A. nodosum* and *E. maxima*, and an NPK-Ref treatment. During this season, there were no significant differences in shoot lengths between the different treatment groups ($P=0.107$) (S4 Fig). In the following season (2022) the average shoot length of three tagged shoots emerging from the fruiting cane was found to be significantly influenced by treatment ($P=0.008$) and time (date of measurement) ($P<0.001$). Specifically, the NPK-treated vines had significantly longer shoots on average compared to water treated vines (EMM: $36.8\pm2.2$ cm vs $29.7\pm2.0$ cm, $P=0.006$) over the 2022 growing season, while no differences were observed for the seaweed extract treated vines relative to control vines (S4 Fig).

For vegetative growth parameters, results showed strong main effects of time (development stage) on the leaf area, leaf dry matter (DM%) and leaf starch content, with less pronounced effects on leaf dry mass (S2 Table). Season showed a strong effect on leaf area, dry mass, and leaf starch content. Treatment only had a significant effect on leaf area ($F_{3,50.9}=5.524$, $P=0.002$), but not on leaf dry mass, specific leaf area, leaf DM% and leaf starch content (S2 Table).

When analysing each season independently, treatment also showed no significant association with leaf dry mass in 2021 ($F_{3,44}=5.340$, $P=0.149$; Fig 2A), and 2022 ($F_{3,44}=7.611$, $P=0.055$; Fig 2B). Leaf area, in contrast, was associated with improvements in both seasons, with vines treated with the *A. nodosum* extract having larger leaves in 2021 (+12%, $P=0.009$; Fig 2C) and 2022 (+15%, $P=0.010$; Fig 2D) compared to control vines. While the effect was stronger (+8%) relative to the NPK-Ref vines, it was not statistically significant ($P=0.355$; Table 1). However, by 2022 leaf area was 18% larger in NPK-Ref vines ($P=0.002$) compared to control vines (Table 1, S3 Table), reaching a comparable size to *A. nodosum* vines ($P=0.905$). While there was no evidence that *E. maxima* increased leaf area relative to control vines in either 2021 or 2022, there were also no significant differences observed relative to *A. nodosum* and NPK-Ref vines in both years (Fig 2C and 2D, Table 1). Overall, leaves were significantly smaller in 2022 compared to 2021 ($F_{1,270}=45.70$, $P<0.001$; S2 Table).

To evaluate the effect of the seaweed extracts on vine physiology, the leaf chlorophyll content index (CCI; proportional to the absolute amount of chlorophyll in the leaf) and the stomatal conductance of the vine were measured. Additionally, fluorescence transients were measured and analysed to obtain the reaction centra density (RC/CS), the maximum quantum yield ($F_v/F_m$), and the electron transport efficiency of photosystem II ($\Phi_{E0}$). The global models showed strong main effects of season and time (development stage) on CCI, RC/CS, $F_v/F_m$, $\Phi_{E0}$, and the stomatal conductance of the vines (S2 Table). A modest effect of treatment was observed only for CCI ($F_{3,44}=3.556$, $P=0.022$), however, strong season × treatment interactions were detected for CCI ($F_{18,689}=10.54$, $P<0.001$) and RC/CS ($F_{18,689}=6.213$, $P<0.001$; S2 Table). The remaining results of the global models are summarised in S2 Table. Given the clear differences between seasons and multiple interactions affecting photosynthetic parameters, separate analyses were done for 2021 and 2022.

In 2021, the main effects of treatment and development stage were associated with differences in CCI. Specifically, between berry set (E-L 27) and the onset of berry softening (E-L 34) the average CCI decreased by 14% from a peak of 18 units to 15.6 ($P<0.001$; Fig 3A). Regarding treatment, *A. nodosum* vines maintained a 12% higher CCI compared to negative control and *E. maxima* vines ($P=0.019$ and $P=0.029$), while compared to NPK-Ref vines there was no significant

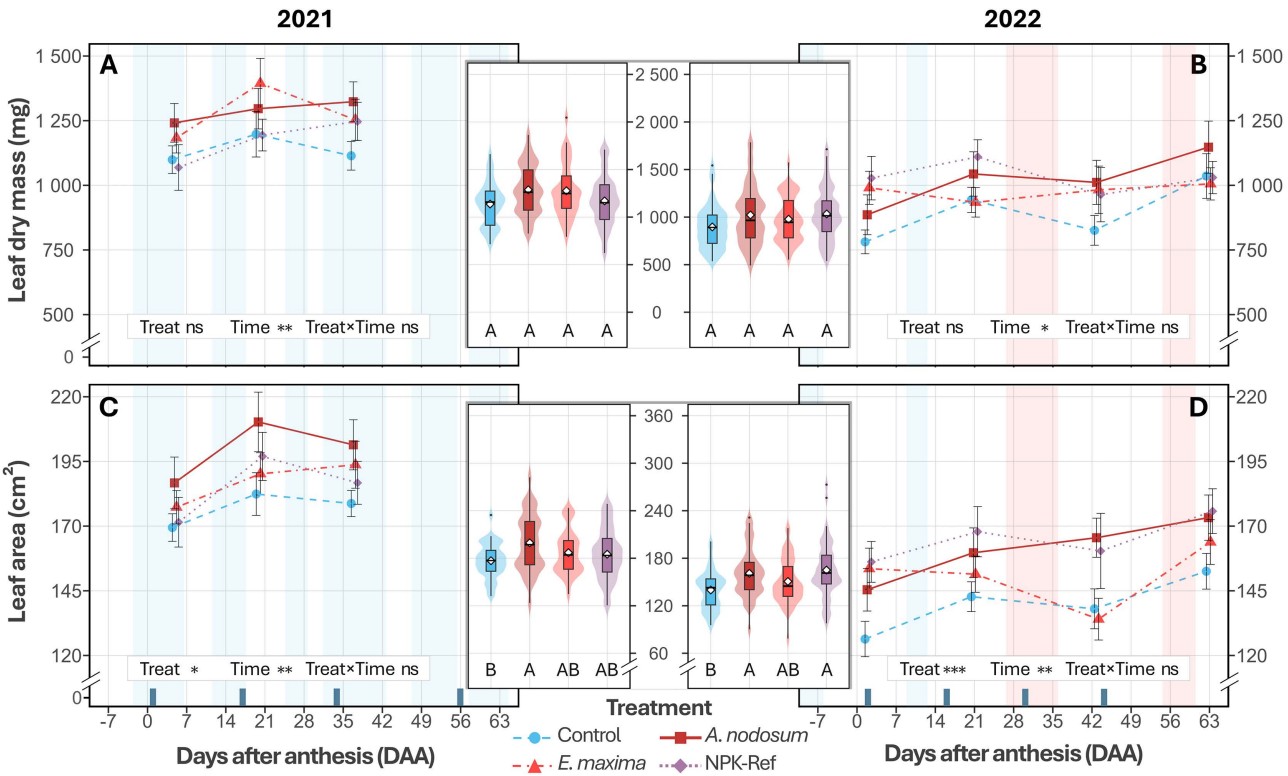

**Fig 2. The evolution of leaf dry mass (2021, A; 2022, B) and leaf surface area (2021, C; 2022, D) of *V. vinifera* cv.** Chardonnay treated with a water control, an *A. nodosum* extract, an *E. maxima* extract, and an NPK-Ref treatment. Each time-series point represents the mean ± standard error (*n* = 12). Violin-boxplots (central panels) show parameter distributions averaged across timepoints per season, with boxplot lines marking the 1st quartile, median, and 3rd quartile, the diamond indicating the mean and whiskers extending to 1.5 × the interquartile range. Different uppercase letters indicate significant differences based on EMMs between treatments during a season, averaged over Time (*P* < 0.05). Treatment application days are shown as dark blue bars on the x-axis (DAA), the first treatment at 18 and 14 days before anthesis is omitted. Samples were taken on dry and clear days as far as possible; blue shading indicates periods with cool/cloudy conditions, and red shading denotes VPD stress (>3 kPa).

difference (*P* = 0.144; Fig 3A, Table 1). In contrast, *E. maxima* vines maintained a comparable CCI to both negative control (*P* = 0.998) and NPK-Ref vines (*P* = 0.892) (Fig 3A, Table 1). Subsequently, in 2022, only the main effect of development stage was associated with differences in CCI, declining steadily from a peak of 13.2 units at E-L 31 to 11.1 units at E-L 35 (-19%, *P* < 0.001; Fig 3B). Furthermore, CCI values were significantly lower compared to 2021 which was characterised by cloudier and cooler conditions (Fig 1A, Fig 3A and 3B).

Similarly, in 2021, treatment and development stage were also significantly associated with reaction centra density. *Ascophyllum nodosum*-treated vines had a 6-7% higher RC/CS compared to control (*P* = 0.009) and *E. maxima*-treated vines (*P* = 0.003), with no difference compared to NPK-Ref vines (*P* = 0.357) (Fig 3C, Table 1). In 2022, however, a significant interaction effect between treatment and development stage (*P* = 0.006) on RC/CS was observed. While most stages showed similar responses across treatments, at véraison (E-L 35), *A. nodosum* vines had a 21% lower RC/CS compared to control vines (*P* = 0.007; Fig 3D, S4 Table), with no significant difference relative to NPK-Ref (*P* = 0.633) or *E. maxima*-treated vines (*P* = 0.260) (Fig 3D, S4 Table). Overall, RC/CS declined by 15% in 2021 and 25% in 2022 over the season (both with *P* < 0.001; Fig 3C and 3D, Table 1). Decreases were particularly sharp in 2022 (-23%, *P* < 0.001) following sustained periods of high vapour pressure deficit (VPD > 3 kPa; Fig 3D) and varied widely between treatments, ranging from -20% (control) to -48% (*A. nodosum*) between E-L 23 and E-L 35 (S4 Table).

**Table 1. Comparison of treatment effects on vegetative growth and physiological parameters of *V. vinifera* cv. Chardonnay averaged over the 2021 and 2022 seasons.**

| Treatment | Leaf area (cm²) | | Leaf dry mass (mg) | | CCI | | RC/CS[1] | | $F_v/F_m$ | | $\Phi_{E0}$ | | Stomatal conductance (mmol m⁻² s⁻¹) | |
|---|---|---|---|---|---|---|---|---|---|---|---|---|---|---|
| **2021** | | | | | | | | | | | | | | |
| Control | 177±4 | b | 1137±39 | a | 16.0±0.3 | b | 4275±52 | b | 0.772±0.003 | ab | 0.310±0.008 | a | 383±13 | a |
| *A. nodosum* | 199±6 | a | 1287±43 | a | 17.9±0.3 | a | 4536±53 | a | 0.777±0.003 | a | 0.326±0.007 | a | 383±13 | a |
| *E. maxima* | 187±5 | ab | 1277±47 | a | 16.1±0.3 | b | 4246±60 | b | 0.770±0.003 | ab | 0.308±0.008 | a | 382±16 | a |
| NPK-Ref | 185±5 | ab | 1170±44 | a | 16.6±0.3 | ab | 4406±65 | ab | 0.766±0.003 | b | 0.305±0.007 | a | 374±15 | a |
| **2022** | | | | | | | | | | | | | | |
| Control | 140±4 | b | 897±33 | a | 11.6±0.1 | a | 3911±59 | | 0.788±0.002 | a | 0.341±0.005 | a | 263±8 | a |
| *A. nodosum* | 161±4 | a | 1022±45 | a | 12.2±0.2 | a | 3950±52 | | 0.793±0.002 | a | 0.351±0.005 | a | 266±9 | a |
| *E. maxima* | 151±4 | ab | 978±34 | a | 12.3±0.2 | a | 4031±57 | | 0.792±0.002 | a | 0.352±0.005 | a | 254±9 | a |
| NPK-Ref | 165±5 | a | 1033±40 | a | 12.2±0.2 | a | 3966±57 | | 0.794±0.002 | a | 0.356±0.005 | a | 253±8 | a |

Data represent mean±standard error (*n*=12 replicate vines per treatment and timepoint), averaged across all measured timepoints for each year. Different superscript letters denote statistically significant differences between treatments within a season (*P*<0.05), based on estimated marginal means (EMMs) with Šidák-adjusted multiple comparisons (α=0.05). Statistical significance was assessed using a Linear Mixed Model (treatment and time as fixed effects; vine as a random effect).

[1]A significant interaction between treatment and phenological stage was observed for RC/CS in 2022; refer to S4 Table for detailed results per timepoint.

Furthermore, *A. nodosum*-treated vines had significantly higher $F_v/F_m$ values on average during the 2021 season compared to NPK-Ref-treated vines (+1.5%, *P*=0.020), but not compared to negative control-treated vines (Fig 3E). This contrasts with 2022, where only development stage (serving as proxy for climatic conditions) was associated with differences in $F_v/F_m$ (*P*<0.001; Fig 3F, Table 1). Meanwhile, *E. maxima* treatments were not associated with significant differences in the average $F_v/F_m$ over either season, compared to control (*P*=0.566) and NPK-Ref vines (Table 1). Notably, however, was an observation following a heat stress event in 2022 at 34 days after anthesis (E-L 32). During this time a maximum temperature of 39.5 °C was reached, while the VPD exceeded 3 kPa for 21 hours over the preceding two days, peaking at 6.7 kPa (Fig 1C). Fourteen days following this event, maximum temperatures remained well below 30 °C and VPD rarely exceeded 2 kPa (Fig 3F, S3 Fig). Vines showed a significant increase in $F_v/F_m$ values compared to the control vines when treated with the *E. maxima* extract (+3.0%; *P*=0.002) and the NPK-Ref treatment (+2.6%; *P*=0.019), while vines treated with the *A. nodosum* extract did not show a significant improvement (S4 Table).

Finally, when evaluating PSII electron transport efficiency $\Phi_{E0}$, neither 2021 nor 2022 showed a significant main effect of treatment on $\Phi_{E0}$ when modelled without interaction (*P*>0.05). However, including a Treat×Time interaction in 2022 (at a large cost to model parsimony, ΔAICc≈230), yielded a borderline significant interaction effect (*P*=0.040). *Post hoc* comparisons indicated modest differences at the final two stages (E-L 34 and E-L 35). At E-L 34, *A. nodosum* vines had a higher $\Phi_{E0}$ than control vines (EMM: 0.398±0.017 vs. 0.352±0.017, *P*=0.033), while *E. maxima* and NPK-Ref were intermediate (S4 Table). By E-L 35, NPK-Ref had the highest $\Phi_{E0}$ (EMM: 0.444±0.017), differing from *A. nodosum* (EMM: 0.390±0.017, *P*=0.009) and *E. maxima* (EMM: 0.399±0.017, *P*=0.038), but not from control (EMM: 0.414±0.017, *P*=0.293). While no consistent overall pattern emerged, results from a repeated-measures correlation analysis (rrm) revealed a significant positive relationship between $F_v/F_m$ and $\Phi_{E0}$ in 2021 (rrm=0.68, *P*<0.001, 95% CI [0.61, 0.75]) and 2022 (rrm=0.75, *P*<0.001, 95% CI [0.71, 0.79]).

## Seaweed extract effect on berry development and quality

The results of the regression analyses showed strong main effects of season and time (development stage) on berry volume, berry mass, berry density and berry DM% (S5 Table). Furthermore, a significant main effect of VPD and an

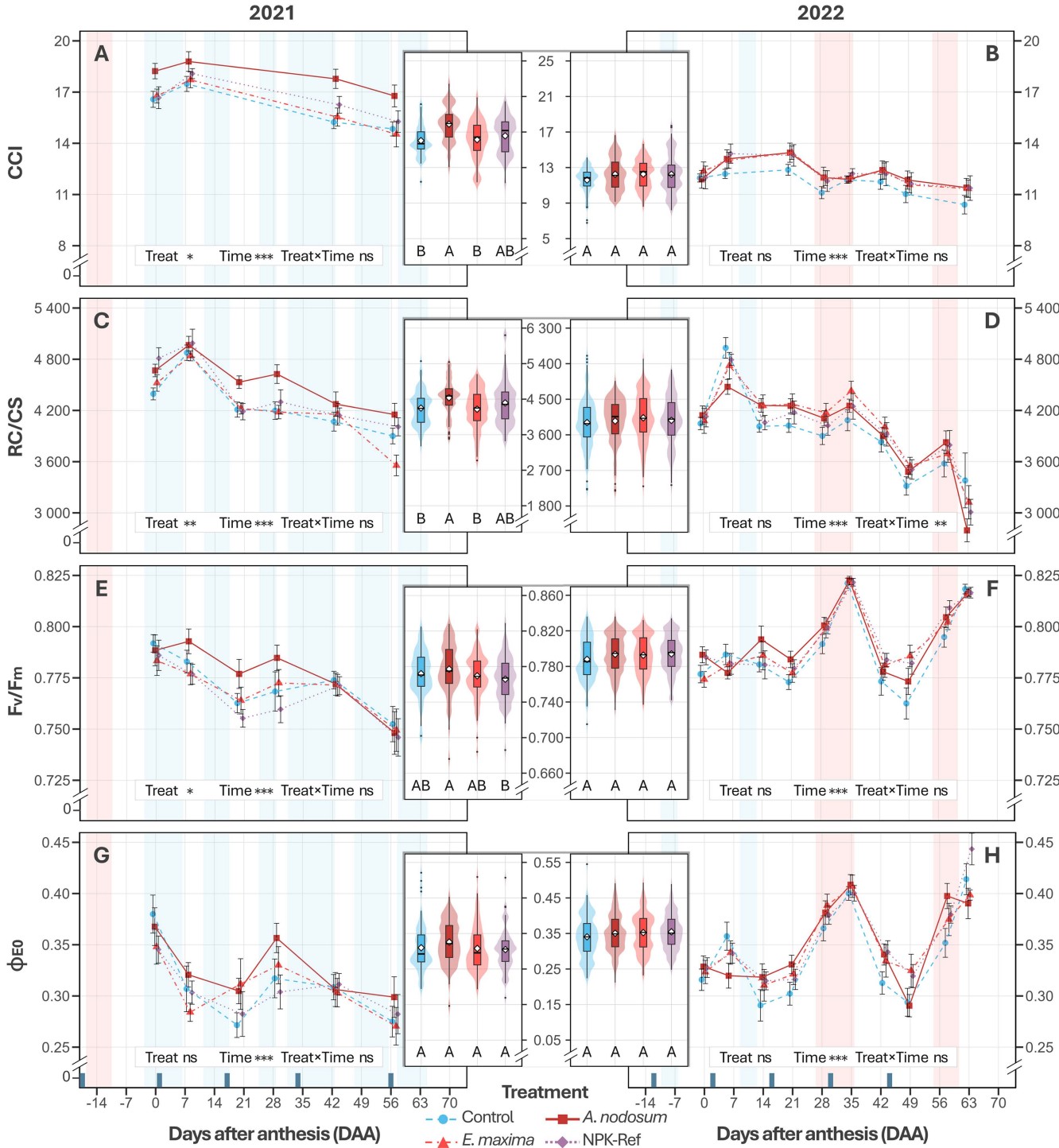

**Fig 3. The evolution of the leaf chlorophyll content index CCI (2021, A; 2022, B) the density of reaction centra per cross section RC/CS (2021, C; 2022, D), the maximum quantum yield $F_v/F_m$ (2021, E; 2022, F) and the electron transport efficiency of photosystem II $\Phi_{E0}$ (2021, G; 2022, H) of *V. vinifera* cv. Chardonnay treated with a water control, an *A. nodosum* extract, an *E. maxima* extract, and an NPK-Ref treatment.** Each time-series point represents the mean ± standard error (*n* = 12). Asterisks indicate significance of the season level factors based on two-way ANOVA (0.01 < *P* ≤ 0.05: *; 0.001 < *P* ≤ 0.01: **; *P* ≤ 0.001: ***; ns = not significant; see S4 Table for means separation by EMMs). Violin-boxplots (central panels) show parameter distributions averaged across timepoints per season, with boxplot lines marking the 1st quartile, median, and 3rd quartile, and whiskers

extending to 1.5 × the interquartile range. Different uppercase letters indicate significant differences based on EMMs between treatments during a season, averaged over Time (*P* < 0.05). When no letters are shown, a significant Treatment × Time interaction was present. Treatment application days are shown as dark blue bars on the x-axis (DAA). Measurements were taken on dry and clear days as far as possible; blue shading indicates periods with cool/cloudy conditions, and red shading denotes VPD stress (>3 kPa).

interaction effect between season and time, was observed for berry density and berry DM% (S5 Table). Treatment did not show a significant association with any of the measured parameters, while interaction terms with treatment were not included in the most parsimonious models (*P* > 0.05; S5 Table). Due to the significant effect of season and the comparatively lower *P*-values, separate analyses per season were also done for berry volume and mass.

Berry volume was not significantly associated with treatment in 2021 (*P* = 0.082), while a treatment effect was observed in 2022 (*P* = 0.005). In particular, compared to control, vines treated with *A. nodosum* and NPK-Ref had an increased berry volume averaged over the season (+8%, *P* = 0.022 and *P* = 0.037), with no significant difference detected between them (*P* = 0.999; Fig 4B). However, at harvest differences were not significant compared to control for either *A. nodosum* or NPK-Ref (*P* = 0.787 and *P* = 0.987; Fig 4A and 4B, Table 2). A similar trend was observed for berry mass, with no

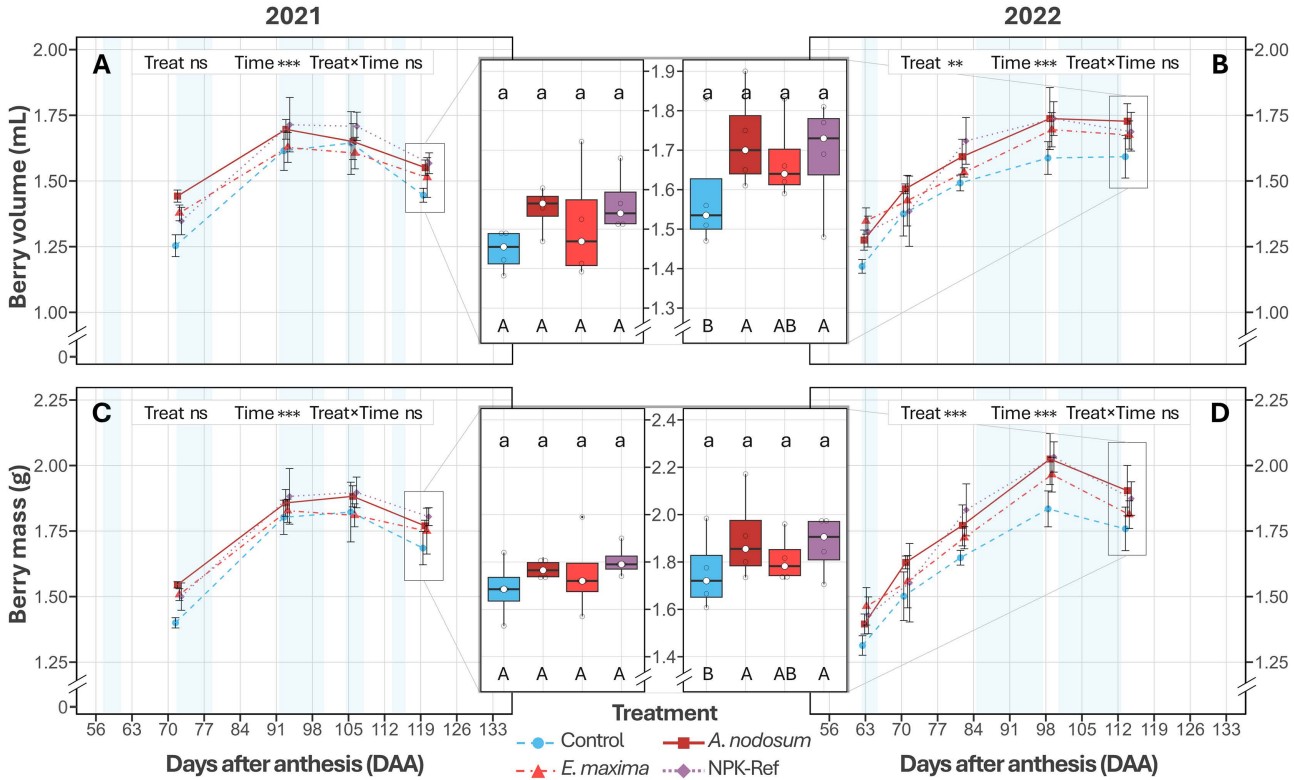

**Fig 4. Berry volume (2021, A; 2022, B) and berry mass (2021, C; 2022, D) of *V. vinifera* cv. Chardonnay treated with water, an *A. nodosum* extract, an *E. maxima* extract, and an NPK-Ref treatment.** Each time-series point represents the mean ± standard error (*n* = 4). Asterisks indicate significance of the season level factors based on two-way ANOVA (0.01 < *P* ≤ 0.05: *; 0.001 < *P* ≤ 0.01: **; *P* ≤ 0.001: ***; ns = not significant). Boxplots (central panels) show parameter distributions at harvest, with boxplot lines marking the 1st quartile, median, and 3rd quartile, and whiskers extending to 1.5 × the interquartile range. Different lowercase letters indicate significant differences between treatments at harvest, based on EMMs (*P* < 0.05). Different uppercase letters indicate significant differences based on EMMs between treatments during a season, averaged over Time (*P* < 0.05). Samples were taken on dry and clear days as far as possible; blue shading indicates periods with cool/cloudy conditions with rainfall.

**Table 2. Comparison of treatment effects at harvest on berry size and quality metrics of *V. vinifera* cv. Chardonnay for the 2021 and 2022 seasons.**

| Treatment | Berry volume (mL) | | Berry mass (g) | | Titratable acidity (g L⁻¹) | | Soluble solids (°Brix) | | Sugar content (mg berry⁻¹) | |
|---|---|---|---|---|---|---|---|---|---|---|
| **2021** | | | | | | | | | | |
| Control | 1.45±0.03 | a | 1.68±0.06 | a | 10.2±0.2 | a | 20.5±0.1 | a | 347±14 | a |
| *A. nodosum* | 1.55±0.03 | a | 1.77±0.02 | a | 10.4±0.1 | a | 20.4±0.3 | a | 361±6 | a |
| *E. maxima* | 1.51±0.08 | a | 1.75±0.09 | a | 10.4±0.2 | a | 20.8±0.4 | a | 366±24 | a |
| NPK-Ref | 1.57±0.04 | a | 1.80±0.03 | a | 10.3±0.2 | a | 21.2±0.6 | a | 383±17 | a |
| **2022** | | | | | | | | | | |
| Control | 1.59±0.08 | b | 1.76±0.08 | a | 7.2±0.4 | a | 21.2±0.2 | a | 373±16 | a |
| *A. nodosum* | 1.73±0.06 | a | 1.90±0.10 | a | 8.3±1.0 | a | 20.7±0.6 | a | 396±25 | a |
| *E. maxima* | 1.68±0.05 | ab | 1.81±0.05 | a | 7.4±0.3 | a | 21.0±0.2 | a | 382±13 | a |
| NPK-Ref | 1.69±0.07 | a | 1.87±0.06 | a | 7.8±0.3 | a | 21.3±0.3 | a | 400±13 | a |

Data represent mean±standard error at the harvest timepoint (*n*=4 replicate panels per treatment). Different superscript letters denote statistically significant differences between treatments at harvest within a season (*P*<0.05), based on EMMs with Šidák-adjusted multiple comparisons (α=0.05). Statistical significance was assessed using a Linear Mixed Model (treatment and time as fixed effects; vine as a random effect).

significant treatment association in 2021 (*P*=0.085), in contrast to 2022 (*P*=0.001), where vines treated with *A. nodosum* and NPK-Ref had on average an increased berry mass (+8%, *P*=0.008 and *P*=0.009; Fig 4D). Again, at harvest, differences compared to control were not significant for either *A. nodosum* or NPK-Ref vines (*P*=0.738 and *P*=0.954; Fig 4C and 4D, Table 2).

Additionally, the berry classical parameters of titratable acidity–TA, pH, total soluble solids–TSS (°Brix), and berry sugar content were measured to evaluate the effect of treatment on ripening dynamics. The global model showed strong main effects of season and development stage (time) on all classical parameters, with a significant season×time interactions for TA, TSS, and berry sugar content. However, neither treatment, nor its interaction terms, had a significant influence in the global models (S5 Table), while the effect of the environmental covariates VPD and rain is shown in S5 Table. Given the significant effect of season, and its interaction with time, subsequent analyses were conducted separately for the 2021 and 2022 datasets to better resolve treatment effects, particularly at harvest.

While treatment was not associated with differences in TA in 2021 ($F_{3,12}$=0.80, *P*=0.519) or 2022 ($F_{3,12}$=0.74, *P*=0.548), the model-estimated mean (EMM) for TA at harvest decreased from 10.3±0.3 g L⁻¹ in 2021 to 7.7±0.2 g L⁻¹ in 2022 (-25%, *P*<0.001; Fig 5A and 5B). The raw means for each treatment are shown in Table 2. In contrast, TSS differed significantly among treatments over the 2021 season (*P*=0.045), but not 2022 (*P*=0.346), while the EMMs for TSS at harvest were comparable in 2021 and 2022 (EMM: 20.7±0.2 and 21.0±0.2 °Brix, *P*=0.999; Fig 5C and 5D). Main effects analysis revealed that *A. nodosum* extract–treated vines had an average increase of 6% in TSS relative to control vines (*P*=0.028; Fig 5C) over the season, and were not different from NPK-Ref vines (*P*=0.999; Fig 5C). However, by harvest, the effect of *A. nodosum* was no longer significant in relation to control vines (*P*=0.999), with the TSS for all treatments having converged (Fig 5C). See Table 2 and S7 Table for raw means.

In the model of sugar content per berry, a significant association with treatment was observed in 2021 (*P*=0.018) and 2022 (*P*=0.004), with no significant interaction with time. Main effects analysis of berry sugar content revealed that, following correction for multiple comparisons, none of the treatments differed significantly during ripening or at harvest, relative to control vines in 2021 (Fig 5E, Table 2). In contrast, for 2022, NPK-Ref vines had a higher sugar content during the season compared to control vines (EMM: 306±9 vs 276±9 mg, *P*=0.012; Fig 5F). The EMM for *A. nodosum* (300±9 mg) did not differ from control vines (*P*=0.099) or NPK-Ref vines (*P*=0.999; Fig 5F). Similarly, *E. maxima* vines did not

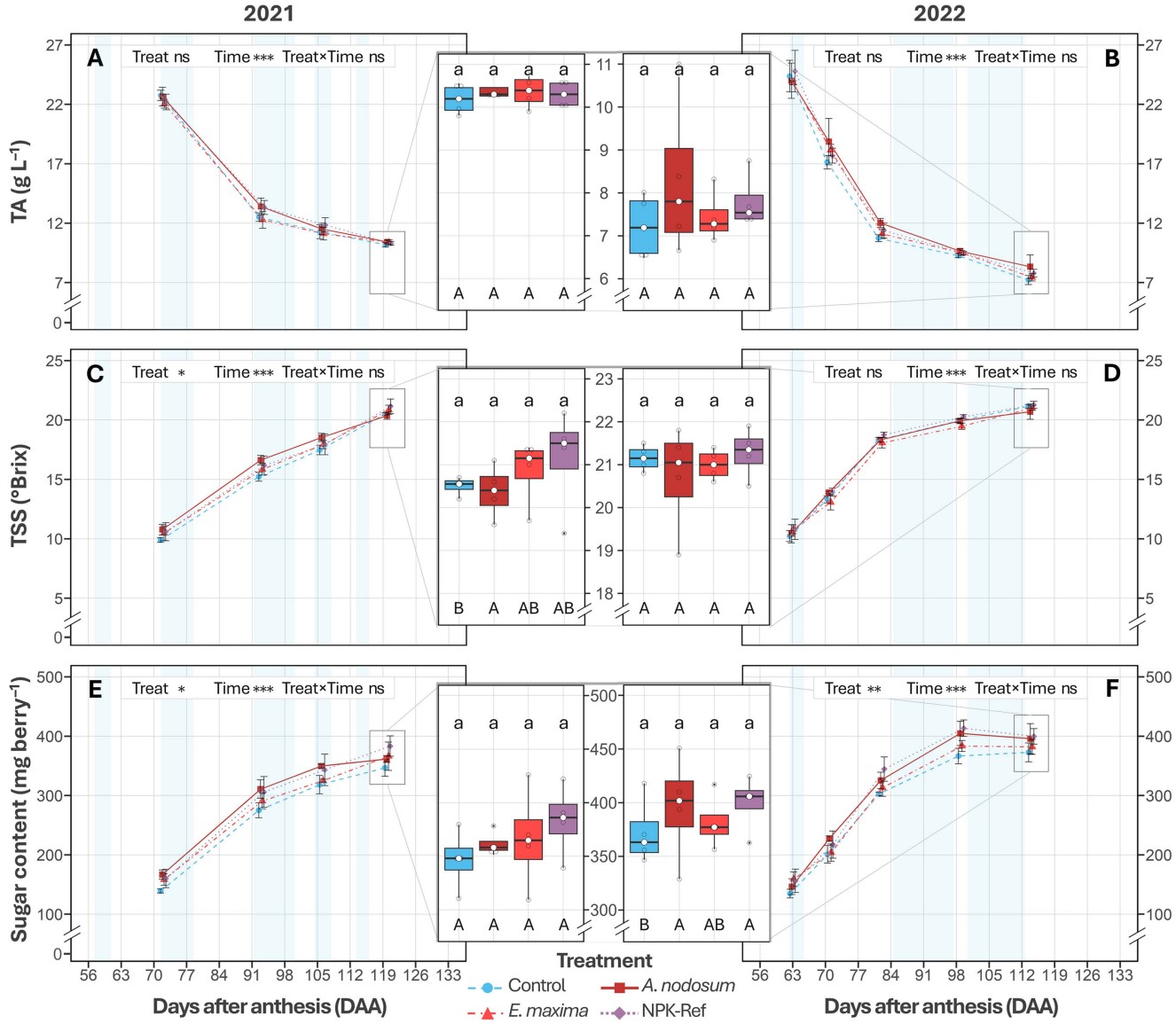

**Fig 5. Development of titratable acidity–TA (2021, A; 2022, B), total soluble solids–TSS (2021, C; 2022, D) and berry sugar content (2021, E; 2022, F) of *V. vinifera* cv.** Chardonnay treated with a water control, an *A. nodosum* extract, an *E. maxima* extract, and an NPK-Ref treatment. Each time-series point represents the mean ± standard error (*n* = 4). Asterisks indicate significance of the season level factors based on two-way ANOVA (0.01 < *P* ≤ 0.05: *; 0.001 < *P* ≤ 0.01: **; *P* ≤ 0.001: ***; ns = not significant. Boxplots (central panels) show parameter distributions at harvest, with box-plot lines marking the 1st quartile, median, and 3rd quartile, and whiskers extending to 1.5 × the interquartile range. Different lowercase letters indicate significant differences between treatments at harvest, based on EMMs (*P* < 0.05). Different uppercase letters indicate significant differences based on EMMs between treatments during a season, averaged over Time (*P* < 0.05). Samples were taken on dry and clear days as far as possible; blue shading indicates periods with cool/cloudy conditions with rainfall.

differ significantly from control or NPK-Ref vines in terms of berry sugar content. Finally, at harvest, EMMs indicated no significant differences between treatments (*P* = 0.531; Fig 5F). The corresponding raw means at harvest ranged from 373 ± 16 mg (control) to 400 ± 13 mg (NPK-Ref); see Table 2 and S7 Table for full values.

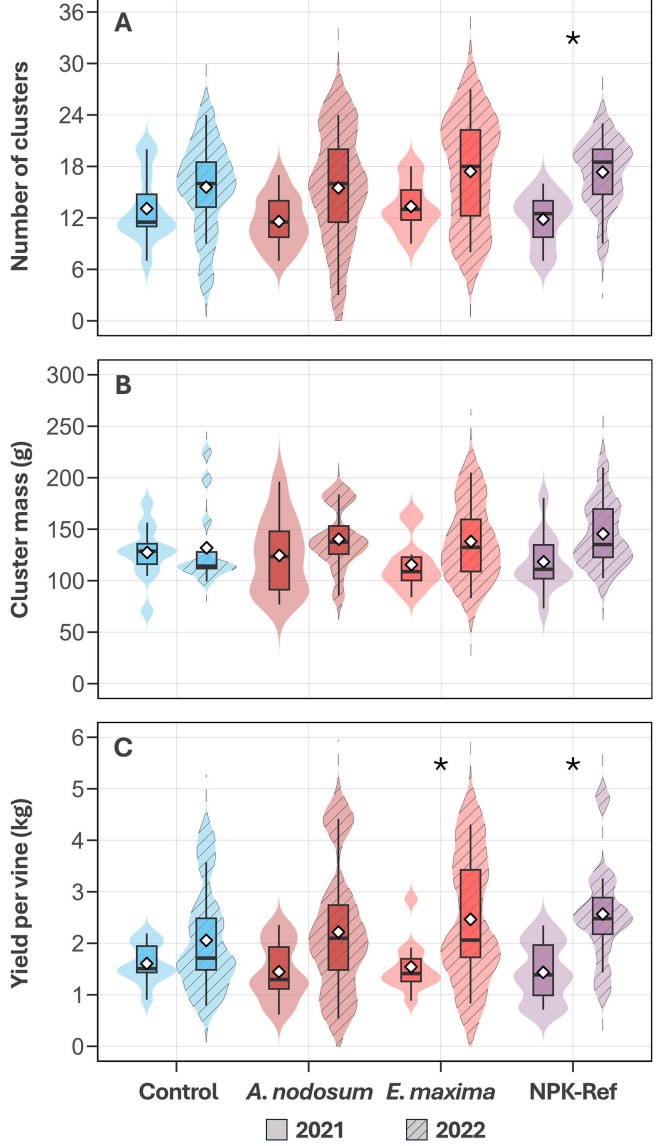

**Fig 6. Total number of clusters (A), average cluster mass (B) and average yield per vine (C) of *V. vinifera* cv.** Chardonnay treated with a water control treatment, an *A. nodosum* extract, an *E. maxima* extract, and an NPK-Ref treatment (*n* = 12) for 2021 (empty violins) and 2022 (striped violins). Samples were taken at technical maturity (average of 21 °Brix). Shaded violin plots show the distribution of data, bar hinges represent quartile values, while whiskers extend to 1.5 × the inter quartile range and the sample mean is indicated with a white diamond. Treatments marked with an asterisk differed significantly from 2021 to 2022 based on their EMMs (*P* < 0.05).

### Seaweed extract effect on harvest parameters

At harvest, the treatments were not significantly associated with differences in the number of berry clusters per vine ($X^2$(1, N = 32) = 2.10, *P* = 0.552; Fig 6A), cluster mass ($F_{3,85}$ = 0.15, *P* = 0.929; Fig 6B) or yield per vine ($F_{3,85}$ = 0.31, *P* = 0.816; Fig 6C), with season being the main driver of differences for these parameters (*P* < 0.001). Although treatments were not associated with differences in yield in 2021 (*P* = 0.820) or 2022 (*P* = 0.652), *a priori* specified comparisons revealed that the increase in yield and number of clusters from 2021 to 2022, varied by treatment. The number of berry clusters was

significantly higher in NPK-Ref vines (+46%, $P = 0.010$), but not in *A. nodosum*, *E. maxima*, nor control vines ($P > 0.05$; Fig 6A). On the other hand, vines treated with *E. maxima* and NPK-Ref had an increased yield per vine of 60% ($P = 0.040$), and 80% ($P = 0.007$) on average respectively, while in *A. nodosum* and control vines there was not a significant increase ($P > 0.05$; Fig 6C). No significant increases in cluster mass were observed for any of the treatments (Fig 6B).

## Discussion

The objective of this study was to evaluate the effects of two commercial seaweed-based biostimulants (based on *Ascophyllum nodosum* and *Ecklonia maxima*) on grapevine vegetative growth and berry quality and ripening dynamics under cool-climate conditions in a Belgian vineyard (Köppen–Geiger Cfb with mean temperature <16 °C). To test our hypothesis—that foliar applications of these extracts have differential effects on vine photosynthetic efficiency, stress resilience, and berry ripening dynamics—we evaluated vine physiological parameters and berry composition over time, as well as yield components over two consecutive growing seasons (2021 and 2022).

### Seaweed extracts show environment-dependent improvements in leaf anatomy and vine physiological parameters

As for many other plant species, grapevine leaves are crucial for photosynthesis, supporting various growth aspects such as new shoot growth and berry ripening [73]. In our study, foliar applications of an *A. nodosum* extract increased leaf area of vines by +12% and +15%, comparable to the NPK-Ref treatment (Fig 2C and 2D). In both seasons, foliar treatments were applied during rain-free periods of at least 24 hours to ensure effective foliar absorption. Similar results have been reported on other white winegrape cultivars such as Narince [74], and white table grape cultivars such as Perlette [42], under warm climate conditions. Enhanced vegetative growth following *A. nodosum* application likely resulted from its rich composition of growth-stimulating phytohormones (cytokinins and auxins), which drive cell division and leaf cell expansion [75,76]. This suggests that *A. nodosum* may be a viable substitute or adjunct to synthetic foliar NPK treatments to increase canopy density. Additionally, *A. nodosum*-treated vines showed increased chlorophyll content index (+12% CCI; Fig 3A), and increased active photosystem II reaction centre density (+6% RC/CS; Fig 3C), under the cooler, cloudier climate conditions in 2021. Betaines present in *A. nodosum* likely modulate the preservation of chlorophyll by delaying chlorophyll degradation and preserving leaf greenness, especially under suboptimal conditions [77]. These effects corresponded with a modest, yet significant increase (+1.5%) in the maximum quantum yield of photosystem II ($F_v/F_m$; Fig 3E) relative to the NPK-Ref treatment, without significant changes PSII electron transport efficiency ($\Phi_{E0}$; Fig 3G). However, in the sunnier 2022 season, these effects were less pronounced, with significantly lower CCI and RC/CS compared to 2021 (Fig 3A and 3B). This is consistent with plant responses involving chlorophyll degradation and activation of their thermal energy-dissipation and antioxidant systems [78,79], to avoid damage to the photosynthetic apparatus under high irradiance. In contrast, *E. maxima* showed limited effects overall, likely due to comparatively higher levels of auxins relative to cytokinins (360 × higher), primarily enhancing root growth and elongation [80]. However, an increase in $F_v/F_m$ was observed two weeks after a heat stress event at the start of véraison in 2022 ($T_{max}$ of 39.5 °C and VPD of 6.7 kPa; Fig 3F, Fig 1A and 1C). Gibberellins and brassinosteroids, present in large amounts in *E. maxima* [39], elicit a wide range of physiological responses—promoting cell division, flower and fruit development, and abiotic stress protection [81]. Brassinosteroids in particular modulate the production of reactive oxygen species (ROS), as well as proline, a key osmolyte with protective effects against abiotic stressors (e.g., oxidative stress, drought, salinity, nutrient limitation, and extreme temperatures) [39,82]. Proline also enhances photosynthesis by protecting RuBisCO activity [83] and stabilising membrane structures to reduce dehydration damage [82]. Although hormonal differences likely explain the observed differential effects between extracts, the hormone concentrations of our specific seaweed batches were not quantified. Differences in nutrient content between *A. nodosum* and *E. maxima* (~59 times more sulphur and ~73 times more magnesium; S1 Table) further support varying physiological effects, given their roles in chlorophyll synthesis and activation of important enzymes like RuBisCO [47]. While

previous studies suggest that *A. nodosum* extracts could maintain plant water status while retaining stomatal opening under water-stressed conditions (allowing for increased carbon assimilation) on some grape varieties [48,50], our study showed no significant differences in stomatal conductance under drier conditions in 2022. This is potentially due to Chardonnay's anisohydric nature, which stabilises photosynthetic performance through lower stomatal sensitivity, at the cost of more variable midday leaf water potentials [84,85]. However, it could be indicative of ample soil moisture due to high rainfall from the previous season, though leaf water potentials were not measured in this study to confirm this hypothesis.

### Berry morphology and classical parameters showed season-dependent responses to seaweed extracts

Berry morphology and ripening were significantly influenced by seasonal variation between 2021 and 2022, with modest treatment effects. No effects were seen in 2021, while in 2022 *A. nodosum*-treated vines had an 8% increase in volume and mass relative to control vines when averaged over the season, similar to the NPK-Ref treatment (Fig 4B and 4D). In contrast, *E. maxima* vines did not differ from either *A. nodosum* and NPK-Ref, or control vines, aligning with a study on three red wine-grape varieties [54]. While leaves were significantly smaller in 2022 compared to 2021 (Fig 2), *A. nodosum* and NPK-Ref vines had relatively larger leaves in each season compared to control vines. Further, more optimal photosynthesis conditions were measured at the start of ripening in 2022 (higher $F_v/F_m$ and PSII electron transport efficiency, $\Phi_{E0}$; Fig 3E to 3H). These factors may have facilitated enhanced carbon assimilation, contributing to increased sugar loading in NPK-Ref and *A. nodosum* vines (306 and 300 mg vs. 276 mg control, Fig 5F). and higher osmotic pressure in berries, promoting water uptake and expansion (Fig 4B) [86]. Previous studies corroborate these findings, showing that *A. nodosum* extracts enhance berry size and mass in white wine- and table grape varieties [42,74,87]. The reported presence of auxins and gibberellins may drive early stage cell division [88], with the comparatively high cytokinin content [38,76] driving later berry expansion [88]. Glycine-betaine, an osmolyte in *A. nodosum*, may further enhance cell expansion through osmotic regulation [77]. The comparatively limited effects of *E. maxima* might be due to its different hormonal balance, notably a higher auxin-to-cytokinin ratio [39]. Additionally, *A. nodosum*'s substantially greater potassium content (≈144-fold higher than *E. maxima*; S1 Table), essential for osmotic balance and water transport, could partly explain its superior performance. Furthermore, when comparing the sugar loading curves for each treatment to the sugar content of the control vines at harvest, it appears that NPK-Ref, *A. nodosum*, and *E. maxima* treated vines may have reached these levels earlier in both seasons (ordered from earliest to latest). These results have implications for wine style, as titratable acidity and pH showed no significant differences between treatments during the season and at harvest, consistent with existing literature [42,44,54,74,87,89–92]. However, by harvest the differences in berry size, mass, TSS, and sugar content were no longer significant. The later half of the ripening period was characterised by increased rainfall (Fig 1B), which may have resulted in a dilution of sugars and acids, alongside an increase in berry volume. However, the lack of significant differences in berry morphology and ripening parameters at harvest likely reflects limited statistical power due to low biological replication (*n*=4), potentially masking small-to-medium effect sizes.

### Yield indicators may be improved by seaweed extract applications

In our study, inter-annual climatic differences were the primary driver of differences ($P<0.001$). While there were no differences in yield components at harvest between treatments within both seasons (Fig 6A to 6C), *a priori* within treatment contrasts between years showed vines treated with *E. maxima* had a significant 60% increase year-on-year, alongside the NPK-Ref vines (+80%). Conversely, the observed yield difference of 53% for *A. nodosum*-treated vines was not significant in this study, likely due to high inter-vine variation in 2022. These findings are consistent with literature, which maintain that yield components, like inflorescence formation and cluster number, contribute 60% to final yield and are heavily influenced by the prior year's environmental conditions and interventions (such as foliar sprays in this study) [93,94]. Treatment effect on yield seemed to be driven mainly by a higher cluster number in 2022 (Fig 6A) combined with a potential increased berry mass (Fig 4C and 4D), from treatment-enhanced photosynthetic capacity (larger leaves, higher CCI, RC/CS and $F_v/F_m$). While previous studies using *A. nodosum* showed significant yield improvements in hot semi-arid climates

following treatment for Chardonnay [52], Thompson seedless [87], Narince [74], and Merlot [91], and in humid subtropical climates for Niágara Rosada [95]. studies in more comparable climate regions showed no significant yield improvements relative to control vines in red winegrape varieties [44,54,89,90].

While our study provides valuable insights into the effects of seaweed extracts on vine photosynthetic performance, influencing berry ripening and yield indicators [96] under cool-climate field conditions, several limitations warrant consideration. First, the study design could be improved with more replicate blocks per treatment for berry classical parameters (increasing $n = 4$ to $n \geq 8$), to improve statistical power to detect medium effect sizes. Next, potential drought stress conditions could be better monitored by quantifying soil moisture content or through evaluating plant water status by pre-dawn leaf water potentials, in particular in 2022 which had drier conditions between E-L 23 and E-L 34. Furthermore, while seaweed extract nutrient content was quantified in 2021, it would have been useful if these measurements were repeated in 2022, while also quantifying the secondary metabolite and hormonal profiles of the extracts. By having a more expansive chemical profile, stronger links can be made between the extract effects under different climate conditions, while also accounting for variation due to storage time of the extracts. Future research could explore the use of individual nutrients or hormones in comparison to the seaweed extracts, to gain a clearer understanding of the modes of action. While previous research comparing foliar and soil irrigation applications have demonstrated that foliar treatments yield the strongest effects [97], further research into the application method is warranted. Specifically, different concentrations and application frequencies, as well as more targeted application timepoints, could provide valuable insights for agronomists [98]. Future research should also investigate the effect of different extraction methods on the composition of the seaweed extracts, and the stability of growth promoting compounds over different storage periods. Finally, although outside of the scope of this study, the eventual wine quality could be impacted by seaweed extracts by, for example, impacting key metabolic pathways during ripening, or altering the composition of the microbiome on the berries and leaves, which could be important aspects for future research.

## Conclusions

This study investigated the effects of *A. nodosum* and *E. maxima* seaweed-based foliar sprays on grapevine growth, berry ripening and yield parameters, compared to a synthetic NPK-Ref solution. The synthetic solution was formulated to match the extracts' NPK levels; however, while nitrogen and potassium matched the levels in *A. nodosum* (the extract richest in these nutrients), phosphorus in the NPK-Ref solution was twice that of *E. maxima* and twenty times that of *A. nodosum*. The study was conducted over two seasons with distinct climatic conditions, in an emerging cool-climate wine region in Belgium. This is a first report demonstrating positive effects of *A. nodosum* on leaf area and photochemical efficiency under cool and cloudy conditions. These changes led to increased berry size and more rapid sugar accumulation during ripening, achieving similar effects to the NPK-Ref treatment. While *E. maxima* did not demonstrate the same effect sizes in terms of leaf growth, it did show potential in helping vines recover from heat stress, however more studies are needed to confirm this. Additionally, it was linked to improved yields in 2022 relative to 2021, similar to the NPK-Ref treatment. A likely reason for the differential effects of the seaweed extracts is the different hormone profiles of the extracts, with *A. nodosum* being more cytokinin dominant compared to *E. maxima* which is more dominant in auxins. Additionally, *A. nodosum* contained significantly more magnesium, potassium, and sulphur, likely enhancing photosynthesis and regulating osmotic balance. This study highlights the potential of seaweed extract as a substitute or adjunct to conventional, synthetic NPK foliar fertilisation, while underscoring the importance of understanding the composition of seaweed extracts from a chemical/metabolic point of view in the context of batch and species level variation.

## Supporting information

**S1 Fig. (A) Overview of the vineyard layout and trial row location from orthophoto-based spatial map of Research Station for Fruit Growing.** Source: National Geographic Institute see S1 Fig. Reference 1. Licensed under CC BY 4.0 (https://creativecommons.org/licenses/by/4.0/), and (B) the pruning and canopy management schedule. (DOCX)

**S2 Fig. Overview of the spraying schedule (A), the experimental randomised block design (B) and the berry sampling schedule (C).** Odd-numbered timepoints represent experimental treatments and even-numbered timepoints the application of a minimal intervention potassium bicarbonate-based pesticide Karma® (A). Note that samples were only taken from the three central vines of each five-vine treatment block to reduce the impact of carryover between treatments (indicated with a check mark). Grey panels and blocks were treated only with Karma® (C).
(DOCX)

**S3 Fig. Comparison of the maximum daily vapour pressure deficit (VPD) (A), and hourly VPD and temperature profiles for the ripening stages of 2021 (B) and 2022 (C).** VPD is shown as a dashed line and temperature as a solid line for both growing seasons (B and C).
(DOCX)

**S4 Fig. Average shoot length of three representative shoots per replicate vine ($n=12$) in 2021 (A) and 2022 (B) of _V. vinifera_ cv.** Chardonnay treated with a water control, an _A. nodosum_ extract, an _E. maxima_ extract, and an NPK-Ref treatment.
(DOCX)

**S1 Table. Overview of the nutrient composition of the applied seaweed extracts in relation to the NPK-Ref treatment.**
(DOCX)

**S2 Table. Effects of treatment, development stage (time), season (2021 vs 2022), their interactions and covariates block and vapour pressure deficit (VPD) on the leaf anatomical and photosynthetic parameters of _V. vinifera_ cv.** Chardonnay over two growing seasons.
(DOCX)

**S3 Table. Average leaf area and leaf dry mass of _V. vinifera_ cv.** Chardonnay in response to treatment with water as control, an _A. nodosum_ extract, an _E. maxima_ extract, and an NPK-Ref treatment. Each value represents the mean±standard error of the raw data ($n=12$). Within each development stage for each season, treatments that showed significantly different responses are indicated with different letters based on their estimated marginal means ($P<0.05$).
(DOCX)

**S4 Table. Average leaf chlorophyll content index (CCI), reaction centra density per cross section (RC/CS), photosystem II (PSII) maximum quantum yield ($F_v/F_m$), PSII electron transport efficiency ($\Phi_{E0}$), and leaf stomatal conductance (σ) of _V. vinifera_ cv.** Chardonnay following treatment with water as control, an _A. nodosum_ extract, an _E. maxima_ extract, and an NPK-reference treatment. Each value represents the mean±standard error of the raw data ($n=12$). Within each development stage for each season, treatments that showed significantly different responses are indicated with different letters based on their estimated marginal means ($P<0.05$).
(DOCX)

**S5 Table. Effects of treatment, development stage (time), season (2021 vs 2022), their interactions and covariates VPD and Rain on berry morphology and classical parameters of V. vinifera cv. Chardonnay .**
(DOCX)

**S6 Table. Average berry volume and berry mass of _V. vinifera_ cv.** Chardonnay in response to treatment with water as control, an _A. nodosum_ extract, an _E. maxima_ extract, and an NPK-reference treatment. Each value represents the mean±standard error of the raw data ($n=4$). Treatments that showed significantly different responses, averaged over the ripening period and at harvest, are indicated with different letters based on their estimated marginal means ($P<0.05$).
(DOCX)

**S7 Table. Average titratable acidity, total soluble solids (°Brix) and sugar content per berry of *V. vinifera* cv.** Chardonnay in response to treatment with water as control, an *A. nodosum* extract, an *E. maxima* extract, and an NPK-reference treatment. Each value represents the mean ± standard error of the raw data ($n = 4$). Treatments that showed significantly different responses, averaged over the ripening period and at harvest, are indicated with different letters based on their estimated marginal means ($P < 0.05$).
(DOCX)

**S1 Data. Data behind all results presented in this manuscript.**
(XLSX)

## Acknowledgments

The authors would like to thank L. Decorte, T. Kocijan, A. Papageorgiou, and A. White for their assistance with the field measurements and F.A.C. van Neerbos for her assistance with the data analysis. Furthermore, the authors thank COMPO Expert GmbH and Acadian Plant Health Ltd. for supplying the seaweed extracts used in this study.

## Author contributions

**Conceptualization:** Johan Yssel, Vicky Everaerts, Wendy Van Hemelrijck, Mathabatha Evodia Setati, Bart Lievens, Erna Blancquaert, Sam Crauwels.

**Data curation:** Johan Yssel.

**Formal analysis:** Johan Yssel.

**Funding acquisition:** Mathabatha Evodia Setati, Bart Lievens, Erna Blancquaert, Sam Crauwels.

**Investigation:** Johan Yssel.

**Methodology:** Johan Yssel, Vicky Everaerts, Wendy Van Hemelrijck, Erna Blancquaert.

**Project administration:** Johan Yssel, Mathabatha Evodia Setati, Bart Lievens, Erna Blancquaert, Sam Crauwels.

**Resources:** Vicky Everaerts, Dany Bylemans, Erna Blancquaert, Sam Crauwels.

**Supervision:** Dany Bylemans, Mathabatha Evodia Setati, Bart Lievens, Erna Blancquaert, Sam Crauwels.

**Visualization:** Johan Yssel.

**Writing – original draft:** Johan Yssel.

**Writing – review & editing:** Johan Yssel, Vicky Everaerts, Wendy Van Hemelrijck, Mathabatha Evodia Setati, Bart Lievens, Erna Blancquaert, Sam Crauwels.

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
