## [Decision Letter · Decision Letter 0]

16 Dec 2024

*Vitis vinifera*

Dear Dr. Crauwels,

While the study compellingly explores the effects of seaweed extracts on grapevines in cool climates, the rationale for focusing on colder regions needs to be better articulated. Simply stating that cold climates represent "another stress factor" is insufficient to justify this focus, particularly as most of the existing literature emphasizes drought and heat stress as the primary targets for seaweed-based treatments. Please revise the introduction to provide a stronger justification.

Additionally, as highlighted by Reviewer #2, although no direct molecular investigations were performed, discussing the potential mechanisms underlying the observed effects—such as phytohormone activity, stress-response pathways, or enhanced nutrient uptake—would greatly enhance the depth of the study and provide a foundation for future research. Furthermore, it would be valuable to explore how these mechanisms might overlap with or differ from those involved in protection against drought or heat stress, as this comparison could offer new insights into the broader applicability of seaweed extracts in stress mitigation.

The term "reproductive parameters" in the title might not be entirely accurate, especially in light of the data and discussion presented. More precise terms such as fruit development, berry attributes, or berry traits would better reflect the study's focus. Please consider revising the title accordingly.

As also suggested by the reviewers, the manuscript would benefit from being less descriptive. For instance, you might consider removing the first sentence of the discussion (line 489) and the sentence in line 292. Additionally, please address the remaining comments and suggestions from the reviewers, as they provide relevant and valuable insights for improving the manuscript.

We look forward to receiving your revised manuscript.

Kind regards,

Hernâni Gerós, PhD

Academic Editor

PLOS ONE

https://journals.plos.org/plosone/s∕File?id=wjVg/PLOSOne_formatting_sample_main_body.pdf and

2. Thank you for stating the following financial disclosure:  [Johan (DJ) Yssel

Project number: 3E200528

KU Leuven Campus Group T

https://www.kuleuven.be/english/campuses/group-t-leuven-campus/Research

The sponsors played no additional roles apart from financing.].  Please state what role the funders took in the study.  If the funders had no role, please state: "The funders had no role in study design, data collection and analysis, decision to publish, or preparation of the manuscript." If this statement is not correct you must amend it as needed. Please include this amended Role of Funder statement in your cover letter; we will change the online submission form on your behalf.

3. We note that your Data Availability Statement is currently as follows: [All relevant data are within the manuscript and its Supporting Information files.] Please confirm at this time whether or not your submission contains all raw data required to replicate the results of your study. Authors must share the “minimal data set” for their submission. PLOS defines the minimal data set to consist of the data required to replicate all study findings reported in the article, as well as related metadata and methods (https://journals.plos.org/plosone/s/data-availability#loc-minimal-data-set-definition ).

Additional Editor Comments (if provided):

Reviewers' comments:

Reviewer's Responses to Questions

**Comments to the Author**

1. Is the manuscript technically sound, and do the data support the conclusions?

Reviewer #1: Partly

Reviewer #2: Yes

2. Has the statistical analysis been performed appropriately and rigorously?

Reviewer #1: Yes

Reviewer #2: Yes

3. Have the authors made all data underlying the findings in their manuscript fully available?

Reviewer #1: Yes

Reviewer #2: Yes

4. Is the manuscript presented in an intelligible fashion and written in standard English?

Reviewer #1: Yes

Reviewer #2: Yes

Reviewer #1: The pdf with the comments that I sent before can be uploaded to editors and co-authors

Reviewer #2: Dear Authors,

I have carefully reviewed your manuscript, which addresses a highly relevant and timely topic with practical implications for viticulture in the face of current climatic crisis. Overall, the manuscript is well-written and logically structured. However, several aspects, particularly related to the Results and Discussion sections, require further attention and improvement. A detailed PDF highlighting specific issues is attached to this report. Still, below, I am outlining the major points for your consideration:

1. Why did the frequency of application differ between years? This aspect is important, especially because you often compare the efficiency of the treatments between 2021 and 2022 growing seasons.

2. In the M&M, you refer that the NPK treatment was performed in order to adjust the nutrient values of the extracts; however, in the discussion, you clearly mention the big differences, regarding nutrient levels, between the two seaweed extracts. So, how was this performed?

3. The manuscript lacks clarity regarding the methodology for post-hoc tests in cases where significant interaction between factors was detected. From the presentation of the figures, it appears that post-hoc tests were conducted uniformly, regardless of the ANOVA results. I am asking this because, as you known, in cases of significant interaction among factors, the simple main effects cannot be easily interpreted, since the response to one factor depends on the other. There are already some programs that can adjust the p-values to allow a fair comparison.

4. I encourage you to avoid overemphasizing non-significant results in the description of the obtained data. Additionally, please consider reorganizing the figures in line with the suggestions provided in the attached PDF.

5. The Discussion section would benefit from a major revision, especially addressing the mechanisms and biological effects underlying the observed treatment responses. Currently, the justification frequently centers on differences in nutrient inputs, but seaweed extracts also contain a diverse range of biologically active compounds that can significantly influence plant responses. I recommend incorporating this aspect into your discussion to provide a more comprehensive analysis.

I hope this revisions help you further improve your MS, so it could be published in Plos One.

**Do you want your identity to be public for this peer review?** For information about this choice, including consent withdrawal, please see our Privacy Policy

Reviewer #1: No

Reviewer #2: **Yes: ** Cristiano Soares

---

## [Author Response · Author response to Decision Letter 1]

31 Mar 2025

We thank the editor and reviewers for their extensive comments and the time they spent reviewing the article. We have incorporated as many of the changes as possible, and where not possible, appropriate reasoning was provided. Please see attached Rebuttal letter, which is divided as follows:

- Response to global comments of the editor

- Summary of major comments and authors' response/correction

- Appendix (with numbered lines) with each individual comment addressed, either briefly or extensively. In the latter case, these were referred in the Summary.

We hope that you find this revised version suitable for publication in PLOS ONE and we look forward to receiving comments in due time.

---

## [Decision Letter · Decision Letter 1]

17 Jun 2025

*Vitis vinifera*

Dear Dr. Crauwels,

Thank you for submitting your manuscript to PLOS ONE. After careful consideration, we believe that it has merit but still requires some minor amendments to meet the quality criteria of PLOS ONE. Both reviewers acknowledged the quality of your revision. Therefore, we invite you to submit a revised version of the manuscript that addresses the remaining points raised during the review process.

Please see the attached documents to access the specific points raised by Reviewer #1.

We look forward to receiving your revised manuscript.

Kind regards,

Hernâni Gerós, PhD

Academic Editor

PLOS ONE

Journal Requirements:

Additional Editor Comments:

Reviewer #1 "minor revision"

Global comment:

The manuscript improved a lot, it’s almost a whole new paper. The authors have addressed the

vast majority of the comments of both reviewers very diligently and competently, with a special

emphasis on Discussion and Statistics, where they did an enormous and great job. I have no doubt

that it’s adequate for publication in PLOS.

I still have some specific comments on few aspects that I’m sure will be easy to handle.

Specific comments (the referred lines are of the revised ms)

(see attached pdf)

Reviewer #2 "minor revision"

Dear Authors,

Thank you for revising your manuscript according to the reviewers’ suggestions. This version shows significant improvement compared to the previous one, particularly in the Results and Discussion sections. Please find below a few minor comments for further adjustment (line numbers relative to the MS file with track-changes ON):

L24 – Remove “-based”

L27 – Use “levels” instead of “level”

L33 – Rephrase to: “improvement of PSII maximum efficiency...”

L35–36 – Please clarify: this increase was in response to which treatment? Ascophyllum nodosum?

L42–43 – Rephrase “cooler climatic conditions in a cool climate region”; it is repetitive.

L98 – Use “focused” rather than “not focused”

L163–165 – Just to confirm: was it not possible to balance P levels between the NPK fertilizer and the extracts?

L173 – Remove “during 2021”

L177–180 – While the rationale for different application periods is clear, the change in application frequency should also be explained.

L802 – Please note that auxins are more commonly associated with root growth; I assume the sentence is the other way around.

L960 – Clarify that the P rate was not the same across treatments.

In addition, I noticed some typographical and grammatical errors throughout the manuscript. I recommend a careful proofreading and a thorough round of language editing to ensure clarity and correctness.

Reviewers' comments:

Reviewer's Responses to Questions

**Comments to the Author**

Reviewer #1: (No Response)

Reviewer #2: All comments have been addressed

2. Is the manuscript technically sound, and do the data support the conclusions?

Reviewer #1: Yes

Reviewer #2: Yes

3. Has the statistical analysis been performed appropriately and rigorously?

Reviewer #1: Yes

Reviewer #2: Yes

4. Have the authors made all data underlying the findings in their manuscript fully available?

Reviewer #1: Yes

Reviewer #2: Yes

5. Is the manuscript presented in an intelligible fashion and written in standard English?

Reviewer #1: Yes

Reviewer #2: Yes

Reviewer #1: see attached pdf

Reviewer #2: see above

**Do you want your identity to be public for this peer review?** For information about this choice, including consent withdrawal, please see our Privacy Policy

Reviewer #1: **Yes: ** Ana Cunha

Reviewer #2: **Yes: ** Cristiano Soares

---

## [Author Response · Author response to Decision Letter 2]

27 Jun 2025

See attached Response to Reviewers_v2 for a detailed overview of the responses to individual comments.

---

## [Decision Letter · Decision Letter 2]

10 Aug 2025

Assessing the potential of seaweed extracts to improve vegetative, physiological and berry quality parameters in *Vitis vinifera* cv. Chardonnay under cool climatic conditions.

PONE-D-24-45452R2

Dear Dr. Crauwels,

We’re pleased to inform you that your manuscript has been judged scientifically suitable for publication and will be formally accepted for publication once it meets all outstanding technical requirements.

Kind regards,

Hernâni Gerós, PhD

Academic Editor

PLOS ONE

Additional Editor Comments (optional):

Reviewers' comments:

Reviewer's Responses to Questions

**Comments to the Author**

Reviewer #1: All comments have been addressed

Reviewer #2: All comments have been addressed

2. Is the manuscript technically sound, and do the data support the conclusions?

Reviewer #1: Yes

Reviewer #2: Yes

3. Has the statistical analysis been performed appropriately and rigorously?

Reviewer #1: (No Response)

Reviewer #2: Yes

4. Have the authors made all data underlying the findings in their manuscript fully available?

Reviewer #1: Yes

Reviewer #2: Yes

5. Is the manuscript presented in an intelligible fashion and written in standard English?

Reviewer #1: Yes

Reviewer #2: Yes

Reviewer #1: The improvement is notorious and most of my specific comments were adequately handle/solved. So, for me, the ms is adequate for publication in PLOS as it stands.

Reviewer #2: Dear Authors,

I am pleased to see that all the points raised by the editor and the reviewers have been properly assessed. In my opinion, the manuscript is now ready for publication. Congratulations on this work!

**Do you want your identity to be public for this peer review?** For information about this choice, including consent withdrawal, please see our Privacy Policy

Reviewer #1: No

Reviewer #2: **Yes: ** Cristiano Soares

---

## [Editor Report · Acceptance letter]

PONE-D-24-45452R2

PLOS ONE

Dear Dr. Crauwels,

I'm pleased to inform you that your manuscript has been deemed suitable for publication in PLOS ONE. Congratulations! Your manuscript is now being handed over to our production team.

Kind regards,

on behalf of

Dr. Hernâni Gerós

Academic Editor

PLOS ONE